# Body mass index and stroke risk among patients with diabetes mellitus in Korea

**Do Kyeong Song, Young Sun Hong, Yeon-Ah Sung, Hyejin Lee** *

Department of Internal Medicine, Ewha Womans University School of Medicine, Seoul, Korea

* hyejinlee@ewha.ac.kr

## Abstract

### Background

Obesity and diabetes mellitus (DM) are both associated with cardiovascular disease. This study aimed to evaluate the association between body mass index (BMI) and stroke risk among patients with DM in Korea since relatively few studies have analyzed this area in detail.

### Methods

We analyzed a total of 56,051 DM patients aged >30 years from the Korean National Health Insurance Service Cohort who had undergone at least one national health examination between 2002 and 2012. BMI scores were divided into six categories, while hazard ratios for stroke were calculated using Cox proportional hazard models.

### Results

Overall stroke risk was positively associated with BMI for both men and women. For ischemic stroke, the risk was positively associated with BMI in women. However, for me, only patients with the highest BMI were at increased risk compared with patients with a BMI of 20–22.4 kg/m². For hemorrhagic stroke, the risk was significantly associated with BMI with a U-shaped association in men. In women, only patients with the lowest BMI had an increased risk of hemorrhagic stroke compared with patients that have a BMI of 20–22.4 kg/m².

### Conclusion

BMI was positively associated with the overall risk of stroke among DM patients in Korea. The risk of ischemic stroke was higher in obese patients compared to overweight or normal-weight patients. However, the risk of hemorrhagic stroke was higher in slimmer patients compared with overweight or obese patients.

**Data Availability Statement:** All relevant data are within the manuscript.

**Funding:** The authors received no specific funding for this work.

**Competing interests:** The authors have declared that no competing interests exist.

## Introduction

Obesity and diabetes mellitus (DM) are both associated with an increased risk of cardiovascular disease, including ischemic heart disease and stroke, which are the leading causes of death [1, 2]. It is also well known that DM is an independent risk factor for cardiovascular disease [3]. People who are overweight or obese are at increased risk of DM. In Korea, the diabetes fact sheet (2020) shows that approximately half of DM patients were considered obese [4]. The risk for cardiovascular events shows a linear association with body mass index (BMI) among Korean adults aged >30 years [5]. Pooled data from 97 prospective cohort studies (after excluding subjects who were younger than 18 years, had a BMI lower than 20 kg/m$^2$, or had a history of cardiovascular disease) showed that being overweight or obese was associated with an increased risk of stroke [6].

However, several studies have suggested the concept of the "obesity paradox," whereby patients with known coronary artery disease, or those who are overweight or obese, actually had a lower risk of mortality compared to those with BMI within the normal range [7], subsequently among in subjects with DM [8, 9]. Furthermore, there are some controversial results regarding the association between BMI and the risk of cardiovascular events in patients with DM. A Swedish study of 13,087 patients aged 30–74 years with type 2 DM showed that being overweight or obese increased the risk of developing cardiovascular disease, including coronary heart disease or stroke [10]. By contrast, there was an inverse association between BMI and the risk of stroke among patients with type 2 DM who were aged 30–94 years in the USA [11, 12]. In a retrospective Korean observational study from 2009 to 2017, which included 249,903 type 2 DM patients who were aged ≥65 years and with no preexisting cardiovascular diseases, the risk of stroke showed an L-shaped association with BMI [13]. In a prospective Korean cohort study, which included 1,338 type 2 DM patients who had suffered from an acute ischemic stroke, the risk of fatal or non-fatal stroke showed an inverse pattern with BMI score after excluding subjects with active cancer or who had died within one month of suffering a stroke [14].

A stroke is a cardiovascular disease that is a great socioeconomic burden. According to statistics from 2018, mortality can be as high as 30 deaths per 100,000 individuals in Korea. The increasing prevalence of obesity was suggested to be one of the reasons why the incidence of strokes among young and middle-aged adults is increasing [15]. However, there have been few studies regarding the association between BMI and stroke risk in DM patients in Korea, including young and middle-aged adults. Therefore, we aimed to evaluate the association between BMI and stroke risk among Korean patients with DM who were aged >30 years and without preexisting cardiovascular diseases using Korean National Health Insurance data.

## Materials and methods

### Data source

We analyzed the cohort database from 2002–2012 obtained from the Korean National Health Insurance Service (NHIS). The Korean NHIS program is a universal health insurance program, which most of the Korean population is enrolled in. Therefore, the cohort can be used as a representation of the general population in Korea, and has been described in detail elsewhere [5]. The participants completed a self-administered questionnaire including information on demographic, medical, and behavioral characteristics.

Informed consent was unnecessary because the data were not collected for the study and the patient records were anonymized before being released by the NHIS. Nevertheless, the study was approved by the Institutional Review Board of Ewha Medical Center, and all methods were performed per the relevant guidelines and regulations [16].

## Study population and outcome variables

Patients with DM aged >30 years who had undergone at least one national health examination between 2002 to 2012 were enrolled in our study. The patient data were obtained from the Korean NHIS cohort. Subjects who had a medical history of stroke, ischemic heart disease, cancer, or chronic obstructive pulmonary disease were excluded based on the self-reported questionnaire or the medical claims data. Subjects who had missing data, which included height, weight, and fasting glucose levels, were excluded (Fig 1).

We calculated BMI as body weight in kilograms divided by height in meters squared (m²). We divided BMI into six categories: <20, 20–22.4, 22.5–24.9, 25–27.4, 27.5–29.9, and ≥30 kg/m². DM was identified based on the questionnaire responses, the medical claims data (ICD code I10-I15 or E10-E14), and the health examination measurements (fasting glucose ≥126 mg/dl). Health behaviors including smoking, alcohol consumption, physical activity, and income were evaluated based on the questionnaire as described in detail elsewhere [5]. The outcome of interest was the time-to-stroke incidence. We defined stroke incidence as the first diagnosis of stroke. Stroke was identified as either ischemic stroke (ICD code I63) or hemorrhagic stroke (ICD codes I60-I62).

## Statistical analyses

Data are shown as both the frequency and proportion for categorical variables. Hazard ratios (HR) for stroke, both ischemic and hemorrhagic, were calculated using the Cox proportional hazards model according to the various BMI. For reference, we used the 20–22.4 kg/m² BMI group for the incidence analysis. All HRs were adjusted for age, smoking, alcohol consumption status, levels of physical activity, income, and family history of cardiovascular diseases. In addition, we conducted a sensitivity analysis after excluding subjects who were diagnosed with stroke <3 years after the baseline measurements to avoid the reverse causality between BMI and stroke development.

We conducted analyses separately by gender. *P* values <0.05 were considered statistically significant. All statistical analyses were performed using SAS (version 9.4, SAS Institute, Cary, NC) or STATA (version 10.0, StataCorp, College Station, Texas).

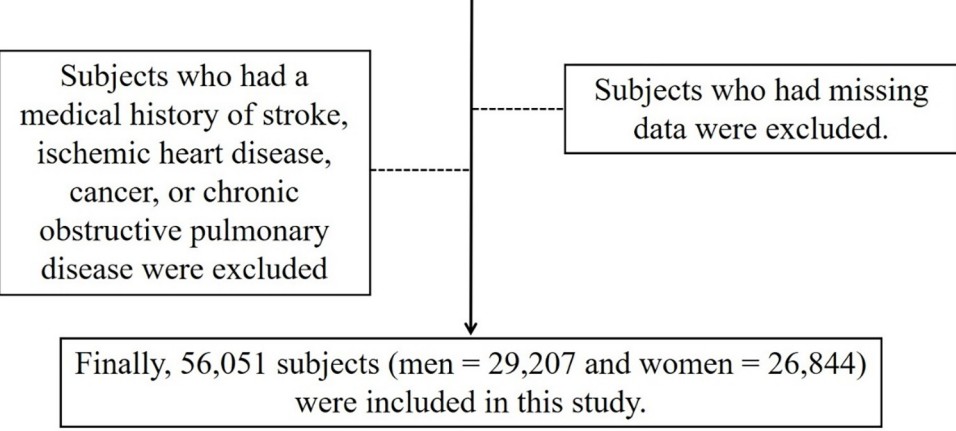

**Fig 1. Flow chart of participant selection.**

## Results

The baseline characteristics of the subjects are shown in Table 1. Altogether, 415,796 subjects (218,388 men and 197,408 women) who were aged >30 years and without a previous medical history of stroke, ischemic heart disease, cancer, or chronic obstructive pulmonary disease were enrolled in the initial analysis. The median follow-up duration was 7.0 years. Within the cohort, 12% of the men and 17% of the women were aged 60 years or older. Subjects with a BMI of <20 kg/m$^2$ comprised 8% of men and 15% of women, while 32.6% of men and 91.4% of women had never smoked. Only 6.4% of men and 7.1% of women had a family history of cardiovascular disease. After exclusions, 56,051 patients with DM (29,207 men and 26,844 women) were included in the final analysis.

Subjects with a BMI score of <20, 20–22.4, 22.5–24.9, 25–27.4, 27.5–29.9, and ≥30 kg/m$^2$ comprised 6.6%, 18.7% (reference), 31.4%, 26.7%, 11.4%, and 5.2% of men with DM, respectively, and 9.0%, 21.2% (reference), 30.0%, 22.2%, 11.1%, and 6.5% of women with DM, respectively. During the study period, 6.30% (n = 1,840) of men with DM and 6.79% (n = 1,822) of women with DM had suffered an ischemic stroke. Additionally, 1.02% (n = 297) of men with DM and 1.06% (n = 285) of women with DM suffered a hemorrhagic stroke. In both men and women, the overall risk of suffering a stroke, either ischemic or hemorrhagic, was positively associated with BMI and was lowest in patients with a BMI of <20 kg/m$^2$ (HR: men 0.97, women 0.94, $p$ value: men 0.735, women 0.399) (Tables 2 and 3). This linear association between BMI and overall risk of stroke was confirmed in a further sensitivity analysis, which excluded patients who were diagnosed with stroke <3 years after the baseline measurements (Tables 4 and 5).

For ischemic stroke, the risk was positively associated with BMI in women with DM. Similar results were obtained in a further sensitivity analysis, which excluded patients diagnosed

**Table 1. Baseline characteristics of the subjects.**

|  | Men (n = 218,388) | Women (n = 197,408) |
|---|---|---|
| Age (years) | | |
| 20–39, n (%) | 92,645 (42.4) | 45,232 (22.9) |
| 40–49, n (%) | 62,200 (28.5) | 78,318 (39.7) |
| 50–59, n (%) | 36,732 (16.8) | 40,564 (20.5) |
| 60–69, n (%) | 20,026 (9.2) | 22,917 (11.6) |
| >70, n (%) | 6,785 (3.1) | 10,377 (5.3) |
| Body mass index (kg/m$^2$) | | |
| <20, n (%) | 17,725 (8.1) | 30,184 (15.3) |
| 20–22.4, n (%) | 49,352 (22.6) | 58,661 (29.7) |
| 22.5–24.9, n (%) | 71,313 (32.7) | 56,999 (28.9) |
| 25–27.4, n (%) | 53,063 (24.3) | 31,946 (16.2) |
| 27.5–29.9, n (%) | 19,371 (8.9) | 13,210 (6.7) |
| ≥30, n (%) | 7,564 (3.4) | 6,408 (3.2) |
| Smoking status | | |
| Current smoker, n (%) | 104,955 (48.1) | 7,575 (3.8) |
| Ex-smoker, n (%) | 21,047 (9.6) | 2,176 (1.1) |
| Never smoker, n (%) | 71,278 (32.6) | 180,390 (91.4) |
| Family history of CVD, n (%) | 13,989 (6.4) | 14,003 (7.1) |
| Hypertension, n (%) | 68,111 (31.5) | 51,188 (25.9) |
| Diabetes mellitus, n (%) | 29,207 (13.4) | 26,844 (13.6) |

CVD: cardiovascular disease.

**Table 2. Hazard ratios for stroke risk by different categories of body mass index among men with diabetes mellitus.**

| | Body mass index (kg/m²) | | | | | |
|---|---|---|---|---|---|---|
| | <20 (n = 1,929) | 20–22.4 (n = 5,451) | 22.5–24.9 (n = 9,176) | 25–27.4 (n = 7,813) | 27.5–29.9 (n = 3,329) | ≥30 (n = 1,509) |
| HR of stroke incidence (95% CI) | 0.97 (0.84–1.13) | 1 | 1.04 (0.94–1.15) | 1.14* (1.03–1.26) | 1.06 (0.92–1.21) | 1.23* (1.01–1.50) |
| P value | 0.735 | 0.055 | 0.415 | 0.014 | 0.438 | 0.038 |
| HR of ischemic stroke incidence (95% CI) | 0.92 (0.75–1.12) | 1 | 0.97 (0.85–1.10) | 1.09 (0.95–1.25) | 1.08 (0.90–1.29) | 1.38* (1.07–1.78) |
| P value | 0.405 | 0.040 | 0.645 | 0.215 | 0.390 | 0.015 |
| HR of hemorrhagic stroke incidence (95% CI) | 1.00 (0.64–1.55) | 1 | 0.74 (0.55–1.02) | 0.70* (0.50–0.98) | 0.89 (0.58–1.36) | 0.99 (0.54–1.84) |
| P value | 0.992 | 0.235 | 0.064 | 0.036 | 0.585 | 0.978 |

HRs were adjusted for gender, health behaviors (smoking, alcohol consumption, physical activity), income, and family history of cardiovascular disease.

* $P < 0.05$. HR: hazard ratio; CI: confidence interval.

with stroke <3 years after the baseline measurements (Tables 3 and 5). However, only men with DM and the highest BMI (HR 1.38, *p* value 0.015) were at increased risk of ischemic stroke compared to patients with a BMI of 20–22.4 kg/m². There was no statistical significance in the sensitivity analysis (Tables 2 and 4).

For hemorrhagic stroke, the risk was significantly associated with BMI, with a U-shaped association in men with DM, where patients with a BMI of 25–27.4 kg/m² (HR 0.70, *p* value 0.036) showed the lowest level of risk. Similar results were obtained in a further sensitivity analysis, which excluded patients diagnosed with stroke <3 years after the baseline measurements (Tables 2 and 4). In women with DM, only patients with the lowest BMI levels (HR 1.90, *p* value 0.004) were at increased risk for hemorrhagic stroke compared with patients with a BMI of 20–22.4 kg/m². However, there were no statistically significant differences in the sensitivity analysis (Tables 3 and 5).

## Discussion

Our study analyzed patients with DM who were aged >30 years using the Korean NHIS cohort and found that BMI was positively associated with the overall risk of a stroke. The risk for

**Table 3. Hazard ratios for stroke risk by different categories of body mass index among women with diabetes mellitus.**

| | Body mass index (kg/m²) | | | | | |
|---|---|---|---|---|---|---|
| | <20 (n = 2,421) | 20–22.4 (n = 5,704) | 22.5–24.9 (n = 8,042) | 25–27.4 (n = 5,976) | 27.5–29.9 (n = 2,969) | ≥30 (n = 1,732) |
| HR of stroke incidence (95% CI) | 0.94 (0.80–1.09) | 1 | 1.07 (0.97–1.18) | 1.11 (1.00–1.23) | 1.17* (1.04–1.32) | 1.23* (1.06–1.43) |
| P value | 0.399 | 0.007 | 0.183 | 0.050 | 0.009 | 0.006 |
| HR of ischemic stroke incidence (95% CI) | 0.88 (0.70–1.10) | 1 | 1.12 (0.98–1.29) | 1.18* (1.02–1.37) | 1.29* (1.09–1.52) | 1.47* (1.20–1.79) |
| P value | 0.274 | <0.001 | 0.105 | 0.024 | 0.003 | <0.001 |
| HR of hemorrhagic stroke incidence (95% CI) | 1.90* (1.22–2.95) | 1 | 1.15 (0.81–1.63) | 0.84 (0.57–1.24) | 1.30 (0.85–1.97) | 1.02 (0.58–1.80) |
| P value | 0.004 | 0.009 | 0.427 | 0.373 | 0.224 | 0.941 |

HRs were adjusted for gender, health behaviors (smoking, alcohol consumption, physical activity), income, and family history of cardiovascular disease.

* $P < 0.05$. HR: hazard ratio; CI: confidence interval.

**Table 4. Hazard ratios for stroke risk by different categories of body mass index among men with diabetes mellitus after excluding patients who were diagnosed with a stroke less than three years after the baseline measurements.**

| | Body mass index (kg/m$^2$) | | | | | |
|---|---|---|---|---|---|---|
| | <20 | 20–22.4 | 22.5–24.9 | 25–27.4 | 27.5–29.9 | ≥30 |
| HR of stroke incidence | 0.97 | 1 | 1.08 | 1.15* | 1.06 | 1.29* |
| P value | 0.764 | 0.127 | 0.248 | 0.034 | 0.515 | 0.038 |
| HR of ischemic stroke incidence | 0.83 | 1 | 0.99 | 1.10 | 1.06 | 1.35 |
| P value | 0.166 | 0.146 | 0.922 | 0.296 | 0.605 | 0.075 |
| HR of hemorrhagic stroke incidence | 1.10 | 1 | 0.81 | 0.62* | 1.16 | 1.06 |
| P value | 0.733 | 0.131 | 0.301 | 0.033 | 0.553 | 0.875 |

HRs were adjusted for gender, health behaviors (smoking, alcohol consumption, physical activity), income, and family history of cardiovascular disease.

* $P < 0.05$. HR: hazard ratio.

ischemic stroke was higher in obese patients compared with those who were either overweight or normal weight. However, the risk of hemorrhagic stroke was higher in patients of normal weight than in overweight or obese patients.

Contrary to the results of a previous study, which showed an L-shaped association between BMI and the risk of stroke among Korean type 2 DM patients aged ≥65 years [13], our study showed that there was a linear relationship between BMI and overall stroke risk among DM patients aged >30 years. In line with the present results, an observational study including 13,087 type 2 DM patients aged 30–74 with no history of cardiovascular disease and with no BMI scores <18 kg/m$^2$ in the Swedish National Diabetes Register showed an increased risk of stroke with an increase in BMI after adjustment for age, sex, DM, type of hypoglycemic treatment, and smoking habits [10]. In a prospective cohort study in England, which comprised 10,568 participants with type 2 DM and no known cardiovascular disease at baseline, overweight or obese patients had a higher rate of cardiovascular events than those of normal weight [17]. In a prospective study of 3,708 Finnish type 2 DM patients aged 25–74, higher BMI scores were associated with increased total and cardiovascular mortality [18]. Data from two prospective cohort studies in the USA, which comprised participants with incidental DM who were free of cardiovascular disease or cancer before a DM diagnosis and with BMI scores >18.5 kg/m$^2$ showed that there was a linear relationship between BMI and cardiovascular mortality [19].

**Table 5. Hazard ratios for stroke risk by different categories of body mass index among women with diabetes mellitus after excluding patients who were diagnosed with a stroke less than three years after the baseline measurements.**

| | Body mass index (kg/m$^2$) | | | | | |
|---|---|---|---|---|---|---|
| | <20 | 20–22.4 | 22.5–24.9 | 25–27.4 | 27.5–29.9 | ≥30 |
| HR of stroke incidence | 0.99 | 1 | 1.07 | 1.04 | 1.06 | 1.21* |
| P value | 0.930 | 0.439 | 0.301 | 0.572 | 0.452 | 0.044 |
| HR of ischemic stroke incidence | 0.94 | 1 | 1.14 | 1.18 | 1.27* | 1.52* |
| P value | 0.648 | 0.015 | 0.156 | 0.089 | 0.032 | 0.002 |
| HR of hemorrhagic stroke incidence | 1.65 | 1 | 1.24 | 0.99 | 1.60 | 1.01 |
| P value | 0.117 | 0.307 | 0.368 | 0.971 | 0.093 | 0.974 |

HRs were adjusted for gender, health behaviors (smoking, alcohol consumption, physical activity), income, and family history of cardiovascular disease.

* $P < 0.05$. HR: hazard ratio.

However, studies have shown that results showing an association between BMI and the risk of stroke among DM patients with DM are not consistent. Contrary to the results of our study, many studies have shown results consistent with the concept of the "obesity paradox." In a large retrospective study of 67,086 patients with type 2 DM between the ages of 30 and 94 (40,431 White Americans and 27,113 African Americans), there was a graded inverse association of BMI with the risk of total stroke after excluding subjects with BMI scores $<18.5$ kg/m$^2$ and adjusting for age, sex, and race [11].

Another retrospective observational study in Louisiana comprising 29,554 patients (aged 30–94, 17,143 African Americans and 12,410 White Americans) who had newly diagnosed type 2 DM showed an inverse association between BMI and overall stroke risk after excluding subjects with a history of stroke and coronary heart disease at the diagnosis and adjusting for age, BMI, race, smoking, income, and insurance type [12]. A study of 25,130 Chinese patients with newly diagnosed type 2 DM, who had no history of myocardial infarction, heart failure, or stroke at baseline and had BMI scores $>18.5$ kg/m$^2$ showed that subjects with a normal-range BMI had a greater risk of ischemic stroke compared with subjects who were overweight or obese [20]. The rate of cardiovascular mortality was higher in normal-weight participants than in overweight or obese participants from a pooled analysis of five longitudinal cohort studies of 2,625 adults with newly diagnosed DM after excluding subjects with BMI $<18.5$ kg/m$^2$. However, the association was not statistically significant [8]. In a meta-analysis of two cohort studies of 92,841 participants with type 2 DM, the risk of cardiovascular mortality exhibited a gradual non-linear increase for subjects with BMI $>30$ kg/m$^2$ and a non-linear decrease for subjects with BMI scores of 22–29 kg/m$^2$ [21]. In a Ukrainian study comprising 30,534 men and 58,909 women with type 2 DM, there was a U-shaped association between BMI and cardiovascular mortality after adjusting for age, smoking status, and alcohol consumption [22]. Levels of physical activity and family history of cardiovascular disease, which could influence the incidence of stroke were not adjusted in the aforementioned studies, and none of the participants were Korean [11, 12, 20, 22]. Subjects who had a medical history of illnesses that could cause low body weight—such as stroke, ischemic heart disease, cancer, or chronic obstructive pulmonary disease—were excluded from our study. However, subjects with chronic diseases such as cancer or chronic obstructive pulmonary disease were not excluded from the previous studies [11, 12, 20, 22]. Differences in study design, sample size, follow-up duration, DM duration, age distribution, smoking history, and inclusion or exclusion criteria may contribute to the differences in the results according to the study.

When stroke subtypes were analyzed separately by gender, we observed subtle differences between men and women. It is well known that although the age-specific rate of stroke incidence is higher in men, women have more stroke events due to their longer life expectancy and the higher incidence of stroke events with age. Scant evidence exists of differences in stroke subtypes between the sexes [23]. Although not completely consistent with our findings, one Japanese study did show differences in the association between BMI and the incidence of stroke depending on sex. Low BMI was a risk factor for total stroke and ischemic stroke in men, while high BMI was a risk factor for total stroke in women among a cohort of 12,490 patients aged 40–69 years after excluding subjects with a history of stroke, myocardial infarction, angina pectoris or cancer [24]. A possible reason for the differences in sex is the difference in sex hormones, the use of oral contraceptives that are known to be associated with increased risk of ischemic stroke, and pregnancy (which increases the risk of thrombosis) [23]. Further studies are required regarding pregnancy, menopause, and the use of oral contraceptives in women with DM.

Ischemic stroke is known to be the most common subtype of stroke in the Korean population, followed by intracerebral hemorrhage and subarachnoid hemorrhage [15]. Similar to the

results among the general populace, the number of DM patients with hemorrhagic stroke in Korea was much lower than the number of DM patients with ischemic stroke in our study. However, the association between BMI and the risk of stroke among subjects with DM differed between stroke subtypes. The risk of ischemic stroke was higher in obese patients compared with overweight or normal-weight patients, while the risk for hemorrhagic stroke was higher in normal-weight patients compared with overweight or obese patients. The inconsistency in these results may be caused by the low incidence of hemorrhagic strokes in DM patients. In addition, the different mechanisms of ischemic and hemorrhagic stroke may affect the results regarding the association of BMI and stroke risk in patients with DM. Asian patients are known to be more at risk of DM even if they are of normal weight [25]. It is possible that the genetic profiles of people prone to normal-weight DM might be associated with the different effects of BMI on the incidence of stroke subtypes in subjects with DM [26].

In addition, because BMI cannot depict body composition or body fat distribution, any assessment of obesity based on BMI might be incorrect. Age-related loss of muscle mass could also result in underweight subjects with excess fatty tissue. Further studies are necessary to measure body composition or various parameters representing visceral obesity, such as waist circumference or waist/hip ratio, which are known to be important for metabolic conditions to evaluate the association between obesity and stroke risk in patients with DM.

To the best of our knowledge, the present work is the first large study to evaluate the associations between BMI and the risk of stroke in Korean DM patients with no previous cardiovascular disease. Our sample was sufficiently large because we used NHIS data representing the general Korean population. To reduce the selection bias related to very slim patients with chronic illnesses, we excluded subjects with a history of stroke, ischemic heart disease, cancer, or chronic obstructive pulmonary disease. In addition, we adjusted for age, smoking, alcohol consumption status, levels of physical activity, income, and family history of cardiovascular diseases that can affect the risk of stroke. Because we included subjects from various age groups, the results of our study can be generalized across the country.

There are several limitations to our study. Because we used questionnaire responses, disease codes, or fasting glucose levels to identify patients with DM, we cannot differentiate between type 1 and type 2 DM. Because we did not measure BMI during the follow-up period, we cannot confirm the results according to the change in BMI over time. Additionally, although we adjusted for various confounding factors, residual confounding factors affecting the development of strokes such as dietary factors, hypertension, hypercholesterolemia, or atrial fibrillation might exist. Also, we did not use various parameters representing obesity, as mentioned above. Finally, we cannot verify the mechanism of the association between BMI and the risk of stroke in patients with DM because the study was retrospective.

In conclusion, obesity is positively associated with overall stroke risk, particularly ischemic stroke, among DM patients in Korea. Therefore, active intervention to maintain appropriate body weight is necessary to reduce the risk of stroke among patients with DM. Randomized clinical trials will be necessary to evaluate the risk of stroke depending on changes in body weight or weight-loss therapies. More prospective studies are also needed to clarify the mechanism of the association between BMI and the risk of stroke in patients with DM.

## Author Contributions

**Conceptualization:** Do Kyeong Song, Young Sun Hong, Yeon-Ah Sung, Hyejin Lee.

**Investigation:** Young Sun Hong, Yeon-Ah Sung.

**Project administration:** Do Kyeong Song, Young Sun Hong, Yeon-Ah Sung, Hyejin Lee.

**Resources:** Hyejin Lee.

**Supervision:** Young Sun Hong, Yeon-Ah Sung, Hyejin Lee.

**Writing – original draft:** Do Kyeong Song.

**Writing – review & editing:** Hyejin Lee.

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
