## [Decision Letter · Decision Letter 0]

27 Jul 2022

PONE-D-22-18934Body mass index and stroke risk among patients with diabetes mellitus in KoreaPLOS ONE

Dear Dr. Lee,

Thank you for submitting your manuscript to PLOS ONE. After careful consideration, we feel that it has merit but does not fully meet PLOS ONE’s publication criteria as it currently stands. Therefore, we invite you to submit a revised version of the manuscript that addresses the points raised during the review process. Please submit your revised manuscript by Sep 10 2022 11:59PM. If you will need more time than this to complete your revisions, please reply to this message or contact the journal office at plosone@plos.org. Please include the following items when submitting your revised manuscript:A rebuttal letter that responds to each point raised by the academic editor and reviewer(s). You should upload this letter as a separate file labeled 'Response to Reviewers'.A marked-up copy of your manuscript that highlights changes made to the original version. You should upload this as a separate file labeled 'Revised Manuscript with Track Changes'.An unmarked version of your revised paper without tracked changes. You should upload this as a separate file labeled 'Manuscript'.

We look forward to receiving your revised manuscript.

Kind regards,

Tariq Jamal Siddiqi

Academic Editor

PLOS ONE

Journal Requirements:

Reviewers' comments:

Reviewer's Responses to Questions

**Comments to the Author**

1. Is the manuscript technically sound, and do the data support the conclusions?

Reviewer #1: Yes

2. Has the statistical analysis been performed appropriately and rigorously? 

Reviewer #1: Yes

3. Have the authors made all data underlying the findings in their manuscript fully available?

Reviewer #1: Yes

4. Is the manuscript presented in an intelligible fashion and written in standard English?

Reviewer #1: Yes

5. Review Comments to the Author

Reviewer #1: Song et al. conducted a study on "Body mass index and stroke risk among patients with diabetes mellitus in Korea", in which they have demonstrated that there is a positive relation between BMI and and the risk of total stroke (hemorrhagic and ischemic), especially ischemic, among patients with diabetes mellitus in Korea. In my opinion, this study may be improved by incorporating the following points:

1. In the introduction, line 50 "Overweight or obesity are also established risk factors for developing cardiovascular diseases" has been repeated, please remove it.

2. Please provide a copy of the preliminary questionnaire if possible, used in the cohort in the accompanying supplement.

3. In the methods, please provide a reference if possible for the relevant guidelines and regulations the study was abided by, lines 89-90.

4. The study should mention more detail regarding the analysis technique. The study uses Cox proportional hazards model, more details are required as to why the Cox's model was used instead of any other type of effect size such as "Odds Ratios"

5. Was any software, website or any particular mathematical equation used to calculate the sample size ?

6. Presenting the p-values for the outcomes evaluated in the study in tables 2-5 may comprehensively highlight the significance of the results.

7. In the results section, only proportions (%) have been mentioned for the stroke incidence. Please mention the hazards ratio (HR) along with their p-values for stroke for the respective BMI classes in the results section as-well.

8. Please write the title of "Figure 1" above the figure itself and not at the start

9. Please proofread the manuscript for grammatical and typographical errors once again.

10. According to author, previous studies have been conducted evaluating the association between BMI and the risk of stroke in patients with DM but with contradictory findings. How does the present study distinguishes itself from the previously conducted studies, apart from a greater sample size ? Some more points on as to how the present results would meaningfully improve the accuracy and reliability compared with the predecessors would be better.

11. Please highlight the avenues for future research which have been opened up as a result of this study, or are needed to contextualize and further the findings presented in this study for better management, and prophylaxis of diabetic obese patients who are at a risk of stroke.

6. PLOS authors have the option to publish the peer review history of their article (what does this mean?). If published, this will include your full peer review and any attached files.

Reviewer #1: **Yes: **Ahmed Kamal Siddiqi

---

## [Author Response · Author response to Decision Letter 0]

23 Aug 2022

PONE-D-22-18934

Body mass index and stroke risk among patients with diabetes mellitus in Korea

PLOS ONE

Dear Dr. Lee,

Thank you for submitting your manuscript to PLOS ONE. After careful consideration, we feel that it has merit but does not fully meet PLOS ONE’s publication criteria as it currently stands. Therefore, we invite you to submit a revised version of the manuscript that addresses the points raised during the review process.

We look forward to receiving your revised manuscript.

Kind regards,

Tariq Jamal Siddiqi

Academic Editor

PLOS ONE

Journal Requirements:

and 

→Thank you for your kind comment. We revised the manuscript according to the PLOS ONE's style requirements.

→Thank you for your kind comment. All relevant data are within the paper. In addition, we presented the p-values for the outcomes in tables 2-5.

Reviewers' comments:

Reviewer's Responses to Questions

Comments to the Author

1. Is the manuscript technically sound, and do the data support the conclusions?

Reviewer #1: Yes

2. Has the statistical analysis been performed appropriately and rigorously?

Reviewer #1: Yes

3. Have the authors made all data underlying the findings in their manuscript fully available?

The PLOS Data policy requires authors to make all data underlying the findings described in their manuscript fully available without restriction, with rare exception (please refer to the Data Availability Statement in the manuscript PDF file). The data should be provided as part of the manuscript or its supporting information, or deposited to a public repository. For example, in addition to summary statistics, the data points behind means, medians and variance measures should be available. If there are restrictions on publicly sharing data?e.g. participant privacy or use of data from a third party?those must be specified.

Reviewer #1: Yes

4. Is the manuscript presented in an intelligible fashion and written in standard English?

Reviewer #1: Yes

5. Review Comments to the Author

Reviewer #1: Song et al. conducted a study on "Body mass index and stroke risk among patients with diabetes mellitus in Korea", in which they have demonstrated that there is a positive relation between BMI and and the risk of total stroke (hemorrhagic and ischemic), especially ischemic, among patients with diabetes mellitus in Korea. In my opinion, this study may be improved by incorporating the following points:

1. In the introduction, line 50 "Overweight or obesity are also established risk factors for developing cardiovascular diseases" has been repeated, please remove it.

→ Thanks for your advice. As you suggested, we deleted that sentence.

2. Please provide a copy of the preliminary questionnaire if possible, used in the cohort in the accompanying supplement.

→ We attached the copy of the preliminary questionnaire of cohort database released by the Korean National Health Insurance Service.

3. In the methods, please provide a reference if possible for the relevant guidelines and regulations the study was abided by, lines 89-90.

→ Thank you for your kind comment. As you recommended, we provided the reference [16] about the relevant guidelines and regulations for strengthening the reporting of observational studies in epidemiology.

4. The study should mention more detail regarding the analysis technique. The study uses Cox proportional hazards model, more details are required as to why the Cox's model was used instead of any other type of effect size such as "Odds Ratios"

→ Thank you for your kind comment. Odds ratios are cumulative over an entire study, while hazard ratios represent instantaneous risk over the study time period. As you recommended, we changed the sentences “Incidence of stroke (ischemic or hemorrhagic) was the outcome of interest and was identified as ischemic stroke (ICD code I63) and hemorrhagic stroke (ICD codes I60-I62).” to “The outcome of interest was the time-to-stroke incidence. We defined stroke incidence as the first diagnosis of stroke. Stroke was identified as either ischemic stroke (ICD code I63) or hemorrhagic stroke (ICD codes I60-I62).”, line 102-104.

5. Was any software, website or any particular mathematical equation used to calculate the sample size?

→ We did not use any particular mathematical equation to calculate the sample size. Instead, we used all subjects who had received health checkup during the period were enrolled in the study.

6. Presenting the p-values for the outcomes evaluated in the study in tables 2-5 may comprehensively highlight the significance of the results.

→ Thank you for your kind comment. As you recommended, we presented the p-values for the outcomes in tables 2-5.

7. In the results section, only proportions (%) have been mentioned for the stroke incidence. Please mention the hazards ratio (HR) along with their p-values for stroke for the respective BMI classes in the results section as-well.

→ Thank you for your kind comment. As you recommended, we presented the hazards ratio (HR) along with their p-values for stroke for the respective BMI classes in the results section.

8. Please write the title of "Figure 1" above the figure itself and not at the start

→ When submitting the manuscript, figures are required to be submitted separately. As you recommended, we inserted the figure just below the title.

9. Please proofread the manuscript for grammatical and typographical errors once again.

→ Thank you for your kind comment. As you recommended, we proofreaded the manuscript for grammatical and typographical errors once again.

10. According to author, previous studies have been conducted evaluating the association between BMI and the risk of stroke in patients with DM but with contradictory findings. How does the present study distinguishes itself from the previously conducted studies, apart from a greater sample size ? Some more points on as to how the present results would meaningfully improve the accuracy and reliability compared with the predecessors would be better.

→ Thank you for your kind comment. As you recommended, “Levels of physical activity and family history of cardiovascular disease, which could influence the incidence of stroke were not adjusted in the aforementioned studies, and none of the participants were Korean [11, 12, 20, 22]. Subjects who had a medical history of illnesses that could cause low body weight—such as stroke, ischemic heart disease, cancer, or chronic obstructive pulmonary disease—were excluded from our study. However, subjects with chronic diseases such as cancer or chronic obstructive pulmonary disease were not excluded from the previous studies [11, 12, 20, 22].” were inserted in the discussion, line 193-199.

11. Please highlight the avenues for future research which have been opened up as a result of this study, or are needed to contextualize and further the findings presented in this study for better management, and prophylaxis of diabetic obese patients who are at a risk of stroke.

→ Thank you for your kind comment. “Randomized clinical trials will be necessary to evaluate the risk of stroke depending on changes in body weight or weight-loss therapies. More prospective studies are also needed to clarify the mechanism of the association between BMI and the risk of stroke in patients with DM.” were inserted in the discussion, line 257-260.

6. PLOS authors have the option to publish the peer review history of their article (what does this mean?). If published, this will include your full peer review and any attached files.

Do you want your identity to be public for this peer review? For information about this choice, including consent withdrawal, please see our Privacy Policy.

Reviewer #1: Yes: Ahmed Kamal Siddiqi

→ Thanks, we used PACE digital diagnostic tool.

---

## [Decision Letter · Decision Letter 1]

15 Sep 2022

Body mass index and stroke risk among patients with diabetes mellitus in Korea

PONE-D-22-18934R1

Dear Dr. Lee,

We’re pleased to inform you that your manuscript has been judged scientifically suitable for publication and will be formally accepted for publication once it meets all outstanding technical requirements.

Kind regards,

Tariq Jamal Siddiqi

Academic Editor

PLOS ONE

Additional Editor Comments (optional):

Reviewers' comments:

Reviewer's Responses to Questions

**Comments to the Author**

1. If the authors have adequately addressed your comments raised in a previous round of review and you feel that this manuscript is now acceptable for publication, you may indicate that here to bypass the “Comments to the Author” section, enter your conflict of interest statement in the “Confidential to Editor” section, and submit your "Accept" recommendation.

Reviewer #1: All comments have been addressed

Reviewer #2: All comments have been addressed

Reviewer #3: All comments have been addressed

2. Is the manuscript technically sound, and do the data support the conclusions?

Reviewer #1: Yes

Reviewer #2: Yes

Reviewer #3: Yes

3. Has the statistical analysis been performed appropriately and rigorously? 

Reviewer #1: Yes

Reviewer #2: Yes

Reviewer #3: Yes

4. Have the authors made all data underlying the findings in their manuscript fully available?

Reviewer #1: Yes

Reviewer #2: Yes

Reviewer #3: Yes

5. Is the manuscript presented in an intelligible fashion and written in standard English?

Reviewer #1: Yes

Reviewer #2: Yes

Reviewer #3: Yes

6. Review Comments to the Author

Reviewer #1: (No Response)

Reviewer #2: (No Response)

Reviewer #3: (No Response)

7. PLOS authors have the option to publish the peer review history of their article (what does this mean?). If published, this will include your full peer review and any attached files.

Reviewer #1: No

Reviewer #2: No

Reviewer #3: No

---

## [Editor Report · Acceptance letter]

21 Sep 2022

PONE-D-22-18934R1 

Body mass index and stroke risk among patients with diabetes mellitus in Korea 

Dear Dr. Lee:

I'm pleased to inform you that your manuscript has been deemed suitable for publication in PLOS ONE. Congratulations! Your manuscript is now with our production department. 

Kind regards, 

on behalf of

Dr. Tariq Jamal Siddiqi 

Academic Editor

PLOS ONE